# Somatic deficiency causes reproductive parasitism in a fungus

Alexey A. Grum-Grzhimaylo [1,2], Eric Bastiaans [1], Joost van den Heuvel[1], Cristina Berenguer Millanes [1], Alfons J. M. Debets[1] & Duur K. Aanen [1✉]

Some multicellular organisms can fuse because mergers potentially provide mutual benefits. However, experimental evolution in the fungus *Neurospora crassa* has demonstrated that free fusion of mycelia favours cheater lineages, but the mechanism and evolutionary dynamics of this exploitation are unknown. Here we show, paradoxically, that all convergently evolved cheater lineages have similar fusion deficiencies. These mutants are unable to initiate fusion but retain access to wild-type mycelia that fuse with them. This asymmetry reduces cheater-mutant contributions to somatic substrate-bound hyphal networks, but increases representation of their nuclei in the aerial reproductive hyphae. Cheaters only benefit when relatively rare and likely impose genetic load reminiscent of germline senescence. We show that the consequences of somatic fusion can be unequally distributed among fusion partners, with the passive non-fusing partner profiting more. We discuss how our findings may relate to the extensive variation in fusion frequency of fungi found in nature.

[1] Laboratory of Genetics, Wageningen University, Wageningen, The Netherlands. [2] Microbial Ecology Department, NIOO-KNAW, Wageningen, The Netherlands. ✉email: duur.aanen@wur.nl

Cheating is an almost universal problem in social systems that depend on cooperation[1]. For example, in multicellular organisms some cells altruistically support reproduction by genetically related cells[2–6]. Such reproductive division of labor can be exploited by cheating mutant cells that manage to become overrepresented among the reproductive propagules even when higher-level organismal fitness declines. Experimental evolution approaches have shown that cheating mutants can emerge in simple multicellular organisms[7–9], but major open questions on how genetic mechanisms mediate or inhibit the spread of cheating have remained unanswered. Understanding the ubiquity of selection pressure for selfish traits and the social processes, that regulate cheating is of paramount importance to appreciate the evolutionary stability of social systems throughout the levels at which social interactions have evolved. Considerable research has focused on cooperation and conflict in prokaryotes and animal societies, but studies in modular multicellular eukaryotes have hardly been done[10–12]. In this paper, we explore the genetic basis and the operational mechanisms of cheating in a filamentous fungus where mycelia often fuse and cell compartmentalization is limited so that nuclei can migrate across the hyphae.

Filamentous fungi form hyphae that branch and fuse regularly to form a dense, radially growing mycelial network[13–15]. Hyphal fusion is a costly somatic trait that pays off because it makes asexual spore production more efficient[16]. Such spores are produced by aerial hyphae, which differentiate from the mycelium and can thus be considered as reproductive structures supported by a substrate-bound somatic hyphal network that acquires the organic resources for growth. The cells in most fungal mycelia are incompletely compartmentalized so that nuclei can move freely through most if not all parts of the fungal colony's mycelial network, including the germline aerial hyphae[17,18]. Therefore, the cooperating units in a fungal colony are the nuclei, and not the cells as in most other multicellular organisms[17]. Filamentous fungi can also fuse with other colonies, potentially leading to the formation of chimeras. Cooperative nuclei in such chimeras may then provide fitness benefits to genetically unrelated nuclei, but face the risk that those could be non-cooperative, selected to exploit their partner nuclei for selfish reproductive benefits. Such cheaters are known to exist. Experimental evolution of a *Neurospora crassa* wild-type strain, which freely fuses, resulted in the emergence of cheaters coexisting with cooperative variants, but a fusion-deficient mutant appeared to preclude the evolution of cheaters[8]. These cheaters were social parasites[19] of the wild type, having an increased probability to become spores relative to the ancestor. Cheater and cooperative variants could be recognized as colonies producing low and normal amounts of asexual spores, respectively. Cheaters also decreased total spore production in mixed culture with the competitor, consistent with the general expectations outlined above.

In this work, we describe the genetic basis and the mechanism of cheating in *N. crassa*. Using whole-genome sequencing of evolved lines and competition assays between defined mutants, we show that fusion deficiency results in a benefit during asexual spore production at the cost of total spore production. Fusion-deficiency mutations prevent cheaters from initiating fusion, but nevertheless enable them to profit from fusion initiated by wild-type mycelia. This benefit is due to reduced contribution to somatic substrate-bound hyphal networks, but increased representation in the aerial reproductive hyphae. However, at higher frequency of the fusion mutant, the mycelial network becomes increasingly fragmented providing a relative benefit to wild-type rich patches. The frequency-dependence of fitness results in an equilibrium between cheater and wild type.

## Results

**Parallel loss of function of fusion genes among cheaters.** To find putative genes underlying cheating in *N. crassa*, we sequenced the genomes of all identified evolved cheaters and cooperative variants that emerged in our lab cultures, and compared them with their ancestors (Supplementary Table 1 and Supplementary Data 1). We found striking parallel evolution among cheaters. In six of the eight independent free-fusion lines the cheater had acquired unique mutations in the gene *soft* (*so*, NCU02794[20]), of which five lead to loss of function based on in silico predictions (a premature stop-codon, intron-junction mutation, and three frameshift mutations). The cheater of the sixth line had acquired a mutation in *so* leading to an amino acid substitution (L825P). The gene *so* encodes a cytoplasmic protein with unknown molecular function, but has been shown to be involved in cell–cell communication between fusing spore germlings[21]. *so*-knockout mutants display strongly reduced fusion between hyphae and germinating spores compared to the wild type, have reduced spore production, and are female sterile[22]. Cheaters of the other two lines, which did not have mutations in *so*, had mutations in other known fusion genes—*ham-5* and *ham-8* (NCU01789 and NCU02811, respectively). Knockout mutants of those genes showed similar phenotypic effects on fusion as *so* knockouts[23,24]. While *ham-8* had acquired a loss-of-function mutation W277* (premature stop-codon formation), *ham-5* had an intronic change, which would not be identified as an inactivating mutation based on in silico predictions. However, we confirmed that the *ham-5* cheater has the same phenotype as a clean *ham-5* deletion strain, suggesting that this seemingly neutral intronic mutation also induces a loss of function. The probability that the observed mutations affected fusion genes in all eight independently evolved cheaters by chance is exceedingly low (exact binomial test, $P = 3.991e-06$, see "Methods" section). The observed parallelism in loss-of-function mutations in fusion genes therefore indicates that losing the ability to initiate fusion underlies cheating in *N. crassa*.

**Reduced fusion underlies cheating.** We used an isogenic deletion mutant of *so* (Δ*so*, Supplementary Table 2) to test this hypothesis, because cheaters also carried additional mutations. We first tested whether disruption of *so* leads to the cheating phenotype, by performing pair-wise competition experiments between the wild type and Δ*so*. By varying the initial frequency of Δ*so* in the competition mixture, we found that Δ*so* indeed has a competitive benefit over the wild type but only at frequencies up to 30% and a disadvantage at higher frequencies (Fig. 1a and Supplementary Figs. 1, 2, see Supplementary Discussion). This result is consistent with the evolution experiment, where cheaters did not go to fixation in seven of the eight lines. Consistent with the characterization as a cheater, total spore production of the mixture decreased when Δ*so* frequency increased (Fig. 1b), corroborating that *so* disruption is causal to the cheating phenotype. To further investigate the link between reduced fusion and cheating we tested seven other known fusion mutants (Δ*mak-1*, Δ*ham-3*, Δ*ham-4*, Δ*ham-5*, Δ*ham-6*, Δ*ham-7*, and Δ*ham-8*; Supplementary Figs. 3 and 4). All of these had a negative effect on spore yield in monoculture, and four of them (Δ*ham-4*, Δ*ham-5*, Δ*ham-6*, and Δ*ham-7*) also a competitive benefit at low frequency, thus displaying similar cheating behavior as Δ*so* (Supplementary Figs. 5 and 6). These results further supported a link between reduced fusion and cheating, although the finding that three of the additional seven tested mutants did not have a competitive benefit shows that there are extra factors involved than fusion ability per se.

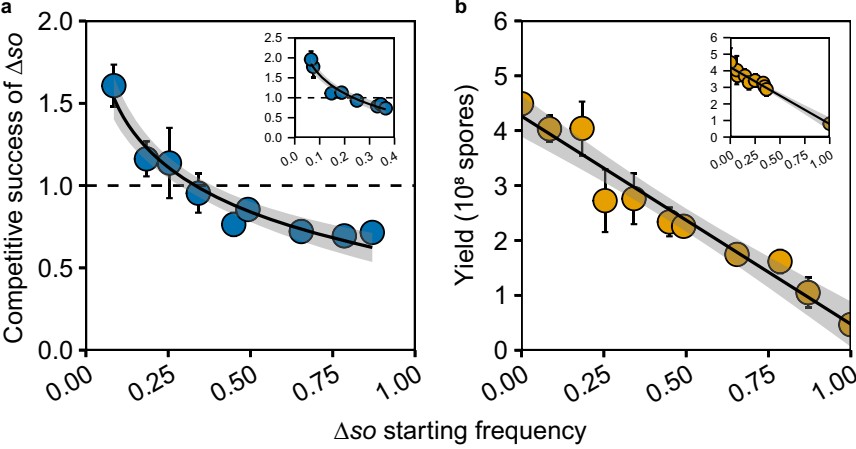

**Fig. 1 Negative frequency dependence of the benefit of Δso in competition with wild type and the negative effect of Δso on sporulation. a** In pair-wise competitions with the wild type, Δso has a competitive advantage relative to wild type at frequencies below ~30%. The competitive success of Δso was calculated using phenotype counts after 4 days as the frequency of Δso after the competition divided by the initial frequency of Δso. **b** As the frequency of Δso spores in the competition mixture increases total spore production decreases. The inserts in **a** and **b** show the results of a separate experiment testing a narrower range of frequencies around 25%. In both panels, shaded areas around the trendlines are 95% confidence areas of the fitted curve. In both panels, error bars are 95% confidence intervals around means calculated from three biological replicates. Source data are provided as a Source Data file.

**Wild-type mycelia initiate fusion with fusion-deficient cheaters.** Since somatic fusion appeared to be a requirement for cheating, it seemed paradoxical that the cheaters had disrupted fusion genes. We therefore hypothesized that it is the wild-type mycelia that fuse with the cheaters, and that the cheaters profit unilaterally. We first set out to test whether wild type could still fuse with the Δso cheater. Earlier studies showed that the Δso mutant is capable of fusion, albeit at very low frequencies[22], but it was still unknown if fusion between wild type and Δso could occur in our experimental setup. We therefore ran transfer experiments with mixtures of Δso and wild type, and measured the fraction of heterokaryotic spores (i.e., containing both Δso and wild-type nuclei) formed by chimeric individuals originating from fusion between wild type and Δso. Our results show that wild type and Δso frequently fuse (Fig. 2a and Supplementary Fig. 7). Already after the first transfer, more than 20% of the spores in Δso/wt mixtures were heterokaryotic, increasing to up to 40% by the fourth transfer. A control experiment where we mixed differently marked Δso revealed almost no heterokaryons (<0.2%). These results demonstrate that Δso cheaters fuse at negligible frequencies with themselves, but wild types frequently fuse with Δso cheaters when Δso is rare, highlighting extreme asymmetry in the fusion trait with the wild type almost always being responsible for fusion with the Δso cheater.

**…. and fusion is necessary for cheating.** Having demonstrated frequent fusion between passive Δso cheaters and active wild types, we then tested whether fusion is required for cheating. To exclude the possibility of fusion we used near-isogenic somatically incompatible wild types as competitors against the Δso cheater. Somatic incompatibility prevents chimera formation by inducing cell death if the nuclei in a fused cytoplasm carry different alleles at somatic-incompatibility loci[25]. This showed that Δso cheaters did not benefit in competition with incompatible wild types, neither at low nor at high frequencies (Fig. 2b and Supplementary Fig. 8). Successful fusion is thus necessary for Δso to become an exploiter of the wild type.

**Cheater nuclei gain advantage during sporulation of a chimera.** We then asked how the Δso cheater can realize a relative benefit within the chimeric mycelium. We hypothesized that hyphae with a high proportion of wild-type nuclei will be more inclined to fuse and thus have a higher tendency to perform supportive somatic functions[16,18]. In contrast, hyphae with more Δso nuclei should have reduced probability to fuse, and will thus have a higher probability to be part of aerial reproductive hyphae that become spores[26]. This hypothesis predicts that the frequency of Δso cheater nuclei should not increase during mycelial growth, and that Δso nuclei should increase their representation in the spores compared to the parental mycelium. qPCR analyses of ten randomly selected chimeric individuals with variable frequencies of Δso showed that the frequency of Δso had decreased after colony growth (Fig. 3a, one-tailed Student's $t$-test, $t$-value $= -11.692$, df $= 9$, $P = 4.803\mathrm{e}{-07}$; Supplementary Fig. 9a, see Supplementary Discussion), irrespectively of the starting frequency (linear regression, $F$-statistic $1,8 = 5.309$, df $= 8$, $P = 0.05014$), indicating that the fusion-deficient mutant does not bias its representation during somatic growth of a chimera. However, we found that Δso nuclei become overrepresented in the spores, provided the frequency of Δso in the chimeric mycelium remained below 60%. Conversely, Δso nuclei were underrepresented in the spores when they reached high frequencies in the chimeras (Fig. 3b, linear regression, $F$-statistic $1,18 = 15.71$, df $= 18$, $P = 0.0009106$; Supplementary Fig. 9b, see Supplementary Discussion). These combined results confirmed that active hyphal fusion is indeed linked with supportive somatic functions, and that fusion-deficient mutants exploit this asymmetry to obtain a relative advantage in the germline. This benefit was negatively frequency dependent, consistent with the finding that cheaters and wild types stably coexisted in all but one of the evolution lines of _N. crassa_[8].

**Discussion**

Figure 4 presents a synthesis of our results, illustrating why cheating is negatively frequency dependent, and resolves the paradox that a fusion-deficient mutant has a relative benefit over a freely-fusing competitor in spite of fusion being required to reap this benefit. While it has been shown before that the costs and benefits of fusion can be asymmetrically distributed between fusing partners[8,27,28], our present results imply that the very ability to fuse is a cause of asymmetrical distribution of these costs and benefits. The fusion-deficient mutants benefit from

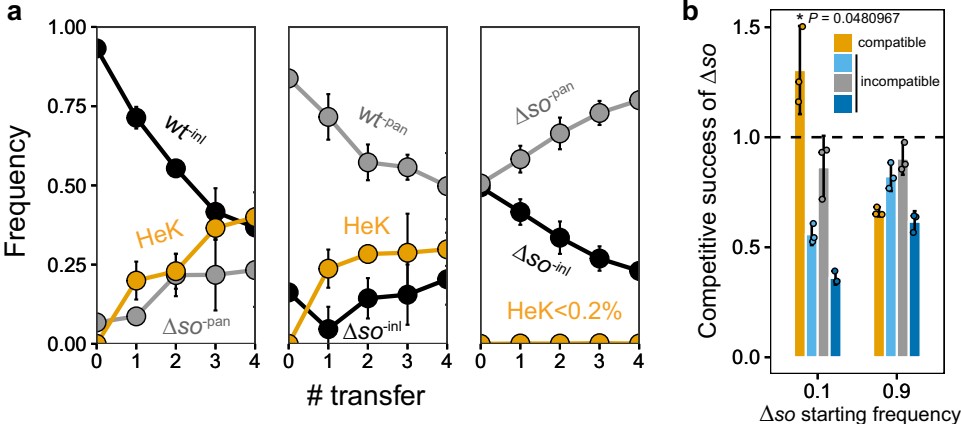

**Fig. 2 The frequency of fusion between wild type and Δso mycelia and the necessity of successful fusion for Δso to exploit the wild type. a** Fusion readily occured between wild type and Δso (first two panels), but hardly between Δso strains (the third panel). The first two panels show the results of transfer experiments with reciprocally labeled Δso and wild-type strains, at low starting Δso frequency (5–15%), while a third panel depicts a control experiment with separately labeled Δso mixed in a 1:1 initial ratio. The spore transfers were performed every 4 days with 1% of total spores. The frequency of genotypes was calculated using phenotype counts at every transfer on non-supplemented plates and supplemented with inositol or pantothenic acid. *inl*-deficient and *pan*-deficient strains are indicated in black and gray, respectively. The orange line shows the frequency of heterokaryons (HeK). **b** Competition experiments between Δso and compatible (orange bars) and three nearly-isogenic incompatible wild-type strains at two starting frequencies of Δso (10 and 90%). Δso frequency only increased when grown with a compatible competitor and only at low frequencies (asterisk marks significant difference of $P < 0.05$; one-tailed Student's *t*-test, *t*-value = 2.986799, df = 2, $P = 0.0480968$). The competitive success of Δso was measured by qPCR. In both panels, error bars are 95% confidence intervals around means calculated from three biological replicates. The replicate data points are overlaid as dots. Source data are provided as a Source Data file.

wild-type strains initiating fusion while never taking such initiative themselves. This demonstrates that hyphal fusion is an altruistic trait, because it facilitates efficient resource distribution and utilization during somatic growth to the benefit of all aerial hyphae that produce and disperse spores during asexual reproduction[26]. Since cheaters have a lower chance to participate in hyphal fusion, most of fusion occurs between hyphae with many wild-type nuclei. Consequently, less-well connected cheater-rich hyphae have a higher probability to become aerial hyphae and to contribute to spore formation.

However, when cheaters reach higher frequencies the culture becomes increasingly fragmented and cheater mycelia will more often remain unconnected to wild-type hyphae. This reduces total spore yield, and provides a relative benefit to wild-type patches that remain isolated from Δso. The frequency dependence of the benefit of the cheater mutant is thus due to the dual effect of fusion deficiency: it provides a benefit at low frequency in competition with a fusion-proficient strain, but causes more fragmentation of the mycelium at higher frequency. Increased fragmentation means more population structure, enabling a shift in the level of selection from within to between mycelia. Selection at the level of mycelia favors the wild type, since wild-type rich mycelia produce more spores.

Interestingly, a recent paper comparing two modes of multicellular development in yeast found analogous results[29]. When grown separately, aggregative (flocculating) yeasts were superior to clonally-developed ("snowflake") yeasts. However, in mixed culture the "snowflake" yeast outcompeted the aggregative genotype. Apparently, by not contributing to the aggregation, but yet being incorporated in the aggregates formed by the flocculating type, the non-aggregating yeast managed to gain advantage. Fusion of *Neurospora* individuals is analogous to aggregation of yeast cells, and, similarly, fusion-deficient cheaters gain a benefit by not expressing fusion ability, but rather by exploiting the fusion-proficient wild type. However, in contrast to our work, the non-aggregating yeast had a benefit at all frequencies, so the details of aggregation and fusion and the respective consequences are likely different.

The finding that not all fusion mutants tested behave as cheaters suggests that there are extra determinants that influence the cheating phenotype. For example, different fusion mutants can have pleiotropic deleterious effects on fitness. In fact, we did not find a competitive benefit relative to the wild type for the Δham-8 strain, while loss of function of this gene was identified in one of the evolved cheaters. We hypothesize that this mutant only has a benefit at low frequencies, since the frequency of this mutant in the evolution experiment after 31 transfers was only 15%. This is very close to the 14% starting frequency used in the competition where we found equal competitiveness (Supplementary Fig. 5), suggesting ~15% is the equilibrium frequency for the Δham-8 cheater.

The convergent emergence of cheaters with the same or similar genetic deficiencies in our evolution experiments shows that they arise often and have sufficiently high selection coefficients to proliferate. Our results imply that the morphological mutants frequently found in lab strains of *N. crassa* are most likely products of unintended selection of *so*-cheater variants during routine propagation[30]. In fact, such morphological "defective" mutants were found in another fungus. The laboratory manual for *Fusarium*[31] states that "The use of standard culturing procedures enables degenerate cultural variants to be recognized quickly and discarded". Interestingly, the manual describes sporeless and female-sterile morphotypes that may arise, but "little is known of the mechanisms that underlie the process". The present work provides one possible explanation and indicates that the threat of cheater mutants during clonal growth is likely generic. Maintaining live cultures by serial transfers, instead of using frozen stocks, may thus reduce overall performance, which begs the question how common or rare cheating mutants are in nature. Although the natural ecology of *N. crassa* is not well known, we hypothesize that the high inoculation density of clonally related spores that we used in our evolution experiments may resemble the natural growth conditions of *N. crassa* because it preferentially grows in post-fire habitats[32]. The sexual spores are heat tolerant and germinate upon heat activation.

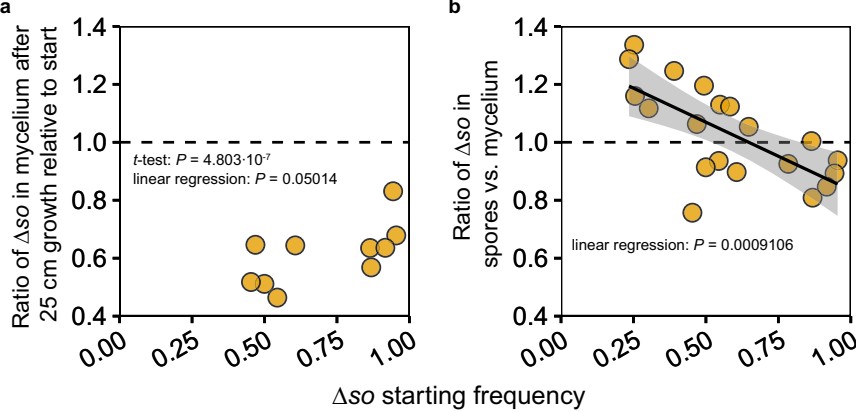

**Fig. 3 How Δso realizes a competitive benefit in chimeras. a** During linear mycelial growth of ten random heterokaryons, the frequency of Δso decreased (all ratios are below 1) (one-tailed Student's *t*-test, *t*-value = −11.692, df = 9, *P* = 4.803e-07), irrespectively of the starting frequency (linear regression, *F*-statistic 1,8 = 5.309, df = 8, *P* = 0.05014). **b** In contrast, during sporulation of these heterokaryons, Δso nuclei have a benefit over wild-type nuclei as long as starting frequencies remain below ~60% (linear regression, *F*-statistic 1,18 = 15.71, df = 18, *P* = 0.0009106). Please note that there are twice as many data points on this panel (compared to **a**) because we used both the end-points and start-points of the same heterokaryon after its 25 cm linear growth for sporulation and the frequencies of Δso nuclei had changed during growth. The shaded area around the trendline is the 95% confidence area. The frequencies Δso in both panels were measured using qPCR. Source data are provided as a Source Data file.

The emerging mycelium then rapidly colonizes empty niche space not only via fast mycelial expansion but likely also via massive local dispersal of asexual spores. If fusion-deficient mutants accumulate during this natural asexual proliferation, it would be reminiscent to germline senescence of an aging multicellular individual that produces fewer offspring with increasing age. Fusion mutants may survive fire events if they end up in homokaryotic sexual spores. However, the germinating spores of fusion-deficient cheaters will remain dependent on fusion with wild-type colonies, which will likely be somatically incompatible because of segregated incompatibility alleles in sexual offspring[25]. We thus predict that genetic load imposed by fusion-deficient mutants rarely persists across seasons in wild *N. crassa* populations.

Specialization on post-fire habitats constrains the asexual life span of *N. crassa* in contrast to many fungi that lack environmentally-restricted life spans. Indeed, some fungal colonies are among the longest-lived organisms as for example *Armillaria* species with an estimated age of at least 2500 years[33,34]. Such extreme longevity requires efficient policing mechanisms against cheating mutants, which may be associated with reduced mutation rates[34,35]. Additionally, many fungi have stronger cell compartmentalization than *N. crassa* and synchronized nuclear division, mechanisms that both restrict dispersal of selfish nuclei through mycelia[17,36]. Restricting fusion via allorecognition has also been recognized as a mechanism to reduce the risk of parasitism[14,37,38]. Reducing fusion frequency itself is yet another way to prevent cheating, and the extensive variation among fungi in fusion frequency, reflected in their tendency to form physically separated mutant sectors within a clonal mycelial colony[39], may thus generally reflect different sensitivities to cheating mutants depending on species-specific ecology. Our experiments demonstrate that fusion is best conceptualized as an altruistic trait that can be exploited. However, the extent of exploitation will depend on the degree to which asexual spores serve local dispersal, which in turn depends on environmental conditions. Furthermore, the ecology may or may not provide extrinsic barriers to the mycelial lifespan, thus limiting the time span for cheater mutations to arise in a single clone, thereby influencing the selective pressure for defense mechanisms.

Buss[17] argued that animals and fungi are more sensitive to exploitation than plants, since plants have rigid cell walls limiting

their mobility, while animal cells and fungal nuclei can disperse within individuals. However, in fungi the opportunities for dispersal also depend on the degree of modular connectivity[18], which does not apply to unitary animals. It seems clear that development of a multicellular organism by non-clonal cell aggregation should give more scope for parasitism than clonal development from a single-celled zygote[1,2,7,40]. However, individuals with clonal development are not immune to social parasitism since nuclei can mutate and be transmitted to other individuals via fusion. Modular organisms such as fungi that lack early germline sequestration are particularly sensitive, since all body parts can ultimately reproduce. However, even early germline sequestration does not guarantee immunity and still leaves the potential for social parasitism via fusion. A striking example from humans came to light decades ago[41] when genetic analysis showed that a woman could not be the biological parent of any of her four presumed children, while her parents were confirmed to be the grand parents of these children. The only reasonable explanation for this enigma was that the mother was a chimera, i.e., the product of a fusion of two full-sister embryos, of which one had contributed the gonads and the other the somatic cells.

## Methods

**Strains and routine subculturing.** The wild-type *Neurospora crassa* strains as well as single-gene knockout mutants (from the *Neurospora* Functional Genomics Project[42]) were obtained from the Fungal Genetics Stock Center (RRID: SCR_008143), Manhattan, Kansas, USA (FGSC[43]). Additional strains carrying *inl* or *pan-2* (or "pan" for short) auxotrophic markers (deficiencies for inositol and pantothenic acid, respectively) in the background of the standard laboratory strain were obtained from ref. [8]. The morphotypes that arose after the evolution experiment in the ref. [8] were used for the whole genome sequencing (see section "Genomes sequencing and analyses"). Supplementary Table 2 lists the strains used in the study (except morphotypes from the evolution lines). For all competitions and transfer experiments we used strains carrying the *mat A* idiomorph. The strains were kept as spore stocks either on silica gel at +4 °C, or in glycerol/peptone (25/7%) suspension at −80 °C. They were routinely propagated on slants with Vogel's minimal medium (VMM) solidified with 2% agar, and containing 2% sucrose as a carbon source[44]. Inositol and/or pantothenic acid (final concentration 50 mg/l) were added to the media to propagate the respective auxotrophic mutants. All incubations were performed at 25 °C with 12/12 h light/dark regime.

**Counting plates.** Counting plates (or "sorbose-VMM") were prepared by adjusting VMM, by substituting sucrose for ʟ(-)sorbose (Calbiochem SDS) with the addition of 0.05% glucose and fructose. This medium keeps the single-spore colonies

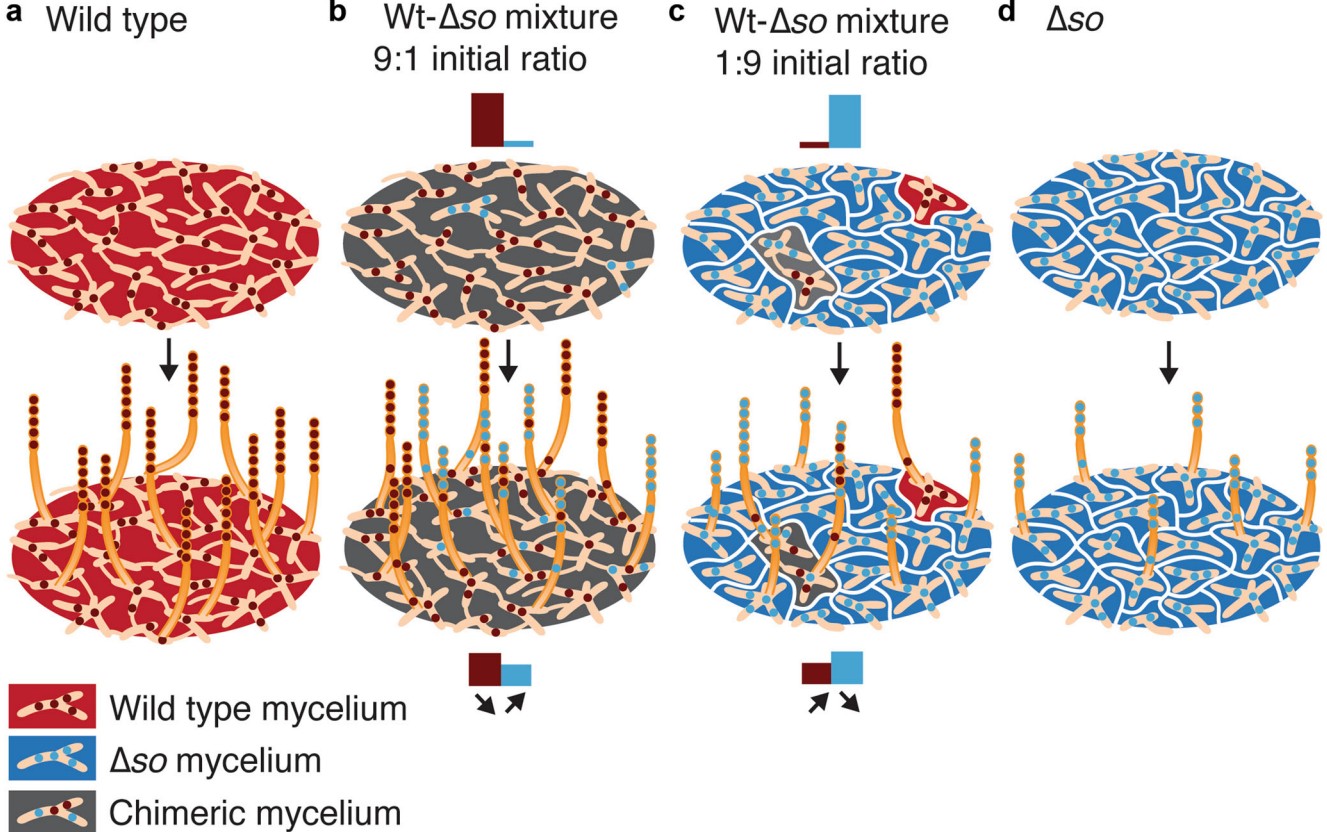

**a** Wild type  **b** Wt-Δ*so* mixture 9:1 initial ratio  **c** Wt-Δ*so* mixture 1:9 initial ratio  **d** Δ*so*

Wild type mycelium
Δ*so* mycelium
Chimeric mycelium

**Fig. 4 Frequency-dependent selection of the Δ*so* cheater and consequences for spore production. a** In a wild-type colony, frequent hyphal fusion allows efficient resource distribution within a mycelial network for optimal development of reproductive aerial hyphae. **b** At low frequency, Δ*so* cheaters are surrounded by connected wild-type hyphae, and have a high chance that at least one wild-type hypha will fuse with them. This gives cheaters access to well-connected wild-type mycelia, and since the cheater nuclei make hyphae less likely to participate in fusion, they have a higher likelihood to become overrepresented in the reproductive aerial hyphae. **c** However, when Δ*so* cheaters reach high frequencies the culture becomes fragmented (white borderlines) and Δ*so* mycelia will often remain unconnected to wild-type hyphae. This reduces total spore yield, and provides a relative benefit to wild-type patches that remain isolated from Δ*so*. **d** A Δ*so* monoculture sporulates very poorly because the somatic structures are maximally fragmented and unable to efficiently mobilize resources to support the reproductive aerial hyphae. The bars indicate how the representation of Δ*so* cheater and wild type change from mycelium to sporulation.

restricted to a small size (~1 cm after a week growth), making feasible counting colonies (up to ~100) on a 9-cm Petri plate. On this medium after 7 days, wild-type colonies produced bright orange heavily-sporulating colonies, while most fusion mutants studied here (Δ*so*, Δ*ham-5*, Δ*ham-6*, Δ*ham-7*, and Δ*ham-8*) produced pale-orange, poorly-sporulating flat colonies. Δ*ham-3*, Δ*ham-4*, produced larger but transparent colonies often with a mycelial rim. Δ*mak-1* fusion mutant formed very small (a few mm) transparent colonies. These phenotypes allowed us to distinguish the mutants from the wild type on counting plates (Supplementary Fig. 4).

**Sexual crosses**. Sexual crosses were performed on slants with synthetic crossing medium (SCM) containing 2% sucrose as a carbon source[44]. As most mutants used in this study were female sterile[45], we used them as males in sexual crosses as follows. Some asexual spores of the knockout mutants were spread onto the 1 d-old recipient female colony. The fertilized culture was incubated until the development of fruit bodies (for 2–3 weeks). The sexual spores were then shot out of the fruit bodies and became visible on the inside of a glass tube as a black film. The sexual spores were collected with a wet cotton swab and were given the required heat shock for 30 min at 61 °C to activate germination, but also to kill the residual asexual spores and mycelial fragments. After the heat shock, the sexual spores were spread onto the counting plates supplemented with hygromycin B (final concentration 200 μg/ml). Since single-gene knockouts were constructed by substituting the gene of interest with a hygromycin cassette, the mutants are resistant to hygromycin, making it a selectable marker[42]. A control plate without hygromycin was used to check whether the re-constituted wild-type strain, which lacks hygromycin resistance, would grow along with the rest of the progeny. The candidate progeny colonies were transferred into slants with VMM, checked for the phenotype, tested for the *inl* or *pan* deficiency, and for the mating type. The mating type was determined by inoculating asexual spores onto 3–5 d-old colony (acted as female) of tester strains (Δ*fl*::*Hyg*r; mat A and Δ*fl*::*Hyg*r; mat a) and checked for the

fruit bodies formation at the inoculum spots. Once the mutant was confirmed and the mating type determined, the glycerol/peptone (25/7%) spore stock was prepared and stored at −80 °C until use.

**DNA extraction**. The morphotypes (i.e., cheaters and social types) of *Neurospora crassa* that emerged after our evolution experiment from ref. [8], as well as the ancestors, were used for full-genome sequencing. To obtain a large amount of mycelium, the strains were grown in 100 ml MY medium (in 500 ml Erlenmeyer's flasks, malt extract—17 g/l, yeast extract 1 g/l) on a rotary shaker (220 rpm) at 25 °C for 1–2 days. A few grams wet weight mycelium was collected and stored frozen at −20 °C. Upon genomic DNA (gDNA) extraction, the mycelium was frozen in liquid nitrogen and ground to powder by a sterile pestle in a sterile mortar. The gDNA was extracted using the CTAB-based protocol as in the ref. [46], with minor modifications (such as reduced mass of starting material to 1 g wet mycelium, and not performing RNase A treatment). The quality and quantity of the gDNA were verified on 0.6% agarose gels stained with ethidium bromide, but also using a Nanodrop 2000 (Thermo Fisher Scientific) and Qubit Fluorometer dsDNA assays (Thermo Fisher Scientific).

For qPCRs analyses, the gDNA from mycelium and spores was isolated with DNeasy Plant Mini Kit (Qiagen) following the manufacturer's instructions.

**Genome sequencing and analyses**. To find mutations, the genomes of the evolved morphotypes (*n* = 36, at ~25× coverage) and the respective ancestors (*n* = 2, at ~40× coverage) of the evolution experiment were sequenced with the Illumina technology (150 bp paired-end reads, HiSeq4000 platform) at BGI (Hong-Kong). The raw reads were filtered and trimmed using the trim-fastq.pl script (--min-length 70 --quality-threshold 30) of the popoolation v.1.2.2 package[47], to ensure only high-quality reads were used for the subsequent mapping. The filtered reads were mapped to the reference genome sequence of *Neurospora crassa* OR74A

(version NC12 + mitochondrial "Supercontig_10.21" contig -> 41102378 nt total with the predicted 9758 protein-coding genes; *Neurospora crassa* Database, RRID: SCR_001372, FungiDB, RRID:SCR_006013) with bwa -mem version 0.7.17-r1188[48]. The SAM files and quality-filtered (mapping quality ≥ 20) sorted BAM files were generated with samtools 1.9[49,50]. The alignment statistics and the average coverage of reads were calculated using "samtools flagstat" and "samtools depth" commands (Supplementary Table 1). Pairwise (ancestor/evolved) mpileup files were generated with "samtools mpileup -Bf" command. Mpileup files were used as inputs for VarScan v.2.4.4[51] to call high-confident SNPs and small indels (<4 bp), using "mpileup2cns --min-coverage 10 --min-var-freq 0.8 --p-value 0.005 --variants --strand-filter 0 --output-vcf 1". The resulting VCF files were inspected for genetic differences by pairwise comparisons, using vcfR v.1.8.0 R package[52], for which we used a minimum allele frequency difference of 0.8 and a minimum coverage of 10 to filter SNPs and indels (Supplementary Data 1). This pipeline did not yield variants for a couple of social variants (genomes of the morphotypes 11t1 and 14t1), hence we treat those genomes as heterokaryotic, and did not analyse them with more relaxed parameters. The detected variants were visually confirmed in the IGV genome browser v. 2.3.94[53,54].

To find larger indels, we used unfiltered BAM files and filtered out all reads with soft and hard clipped reads, as well as indel and deletion calls. Furthermore, we filtered on flags 67, 131, 115, 179, 81, 161, 97, 145, 65, 129, 113, and 177, which potentially indicate large indels or deletions or chromosomal rearrangements. Using a custom R 3.6.1[55] script, we quantified coverage of all these reads in 100 bp windows. The resulting coverage distribution was divided to the total coverage of the initial unfiltered BAM files, which therefore yielded the frequencies of "alternate call" mapped reads to those with "normal" mapping. These frequencies were then compared between all pairwise samples, ordered upon frequency and visually inspected using IGV without any a priori cut-offs. Half of the variants detected using this custom-script method showed the variants which were already detected by VarScan (these are colored red in the mutations list in Supplementary Data 1).

### Calculating the probability of parallel mutations in fusion genes. The probability that the observed mutations affect fusion genes in all eight independently evolved lines can be conservatively calculated as follows. First, cheater morphotype evolved in all eight evolution lines independently with a number of mutations per strain of ~15 (conserved estimate based on Supplementary Data 1). Second, the probability of a mutation hitting a fusion gene will be 75/10,000 given about ~75 known fusion genes and a total number of protein-coding genes in *N. crassa* to be around 10,000. Putting these data together in a binomial test would yield a *P* value of 3.991e−06: The R code line binom.test(8,8*15,75/10,000). To correct this probability for the strains that did not evolve cheater morphotype the same test can be applied as follows. Binom.test(8,8*15,(10,000 − 75)/10000) yielding a *P* value <2.2e−16. This negligible contribution would not affect the overall conclusion that the mutations hitting fusion genes eight times independently is not a chance effect.

### Competition assays. To determine competitive success of fusion mutants against the wild type, we performed pairwise competition assays as follows. The strains were pregrown in VMM slants for 6–7 days, and washed with sterile MQ-water to obtain spore suspensions. The spore concentrations were brought to $4 \times 10^7$ sp/ml and mixed at different ratios (from 10 to 90% of Δ*so* in the mixes with ~10% increments). We performed a separate experiment covering a range 5–40% of Δ*so* with ~5% increments to look at the competition dynamics with the Δ*so* frequency at around 25%. Both experiments included monoculture controls. Fifty microliter of the spore mixes were mat-inoculated in triplicates onto the surface of a slanted VMM medium, left horizontally to let the spore suspension to soak for 15 min, and then incubated upright for 4 days (or for 1–6 days to measure competitiveness in time, see section "No temporal benefit of the Δ*so*-cheater" in Supplementary Discussion). A small quantity of inoculum was spread onto 4–8 counting plates to estimate the realized initial frequency by phenotype counts, and correct for this value later when calculating competitiveness. After the competition, the spores were washed with 5 ml of water (vortexed for ~20 s), diluted and inoculated onto counting plates. On counting plates, the fusion mutants could be distinguished by phenotype from the wild-type colonies (Supplementary Fig. 4), hence by counting each type the end frequency of the mutant could be determined. However, there is a possibility of errors due to stochastic effects on the phenotypes and a chance of chimera formations between the two genotypes masking the phenotype. To verify this phenotypic approach, we analysed the Δ*so*/wild type competitions (with 10 and 90% starting ratios of Δ*so*) using qPCR (Supplementary Fig. 1, see section "Frequency of Δ*so* cheater in heterokaryons"), which yielded conceptually similar results. The discrepancy at low starting frequency of Δ*so* may indicate underestimation of mutant counts due to its recessiveness in a heterokaryon state with the wild type.

For seven other fusion mutants (Δ*mak-1*, Δ*ham-3*, Δ*ham-4*, Δ*ham-5*, Δ*ham-6*, Δ*ham-7*, and Δ*ham-8*), the competitions were done at a single starting frequency (~10%) and analysed by counting as with Δ*so*, since the mutants could be distinguished by phenotype on counting plates (Supplementary Fig. 4). The statistical significance of competitions was tested using one-sample one-tailed Student's *t*-test: Δ*so*/wt: *t*-value = 5.7860, *P* = 0.0142975; Δ*mak-1*/wt: *t*-value = −85.1019, *P* = 0.999931; Δ*ham-3*/wt: *t*-value = −9.6499, *P* = 0.994715; Δ*ham-4*/

wt: *t*-value = 3.7405, *P* = 0.0323107; Δ*ham-5*/wt: *t*-value = 9.5707, *P* = 0.00537052; Δ*ham-6*/wt: *t*-value = 5.7381, *P* = 0.0145270; Δ*ham-7*/wt: *t*-value = 16.1410, *P* = 0.00190842; Δ*ham-8*/wt: *t*-value = −0.2916, *P* = 0.600992).

### Spore yield. Spore yields were determined as in the ref. [8], except that the counting was done mostly with CASY TT Cell Counter (OMNI Life Science & Co KG, Germany). Occasionally though the haemocytometer was used. By counting random spore samples, we verified these two methods produce similar results.

### Heterokaryon formation. To obtain heterokaryotic cultures, the *inl* and *pan* deficient strains were pre-grown separately in VMM (inositol and pantothenic acid supplemented) slants for 6–7 days, to generate sufficient number of spores. The spores were washed off with sterile MQ-water. The *inl* and *pan* deficient spores were mixed in 1:1 ratio and mat-inoculated on fresh VMM (no supplementation) slants to allow only heterokaryotic mycelium to propagate via markers complementation, as a result of the occasional fusion between the deficient strains. The emerged heterokaryotic colony allowed to sporulate (i.e., incubated for 4 days), spores were collected with sterile MQ-water and appropriate dilutions were plated on counting plates (no supplementation) to allow only heterokaryotic spores to germinate. The agar plug with a piece of a colony (~1 mm³) was used in subsequent experiments.

### Frequency of Δso cheater in heterokaryons. To determine Δ*so* cheater dynamics in heterokaryotic mycelium and derived spores, we isolated ten independent heterokaryotic colonies (5 of wt⁻ⁱⁿˡ + Δ*so*⁻ᵖᵃⁿ and 5 wt⁻ᵖᵃⁿ + Δ*so*⁻ⁱⁿˡ), which were randomly picked from non-supplemented counting plates (see section "Heterokaryon formation"), and let them grow in a "race tube" (made of 50-ml disposable pipette, ~25 cm long) filled with ~1 cm wide VMM medium layer. After the mycelium has reached the end, the tubes were cut open, mycelium was collected from the start and the end of the tube and snap-frozen in liquid nitrogen and put to −80 °C for the subsequent DNA extraction (see section "DNA extraction"). Part of this mycelium was put into small glass tubes filled with non-slanted 0.5 ml of VMM. The reasoning behind using non-slanted small tubes was to restrict the mycelial outgrowth from the plug as much as possible. This way, the inoculated mycelium would mostly produce spores, limiting intramycelial allele dynamics upon mycelial outgrowth. After 7 days, the spores were collected with sterile MQ-water, centrifuged and stored at −80 °C for the subsequent DNA extraction (see section "DNA extraction"). This experiment was performed both in VMM supplemented with inositol and pantothenic acid, and in non-supplemented VMM using the same ten independent heterokaryons, to check how the heterokaryon enforcement would affect the cheater's dynamics (see section "Heterokaryon enforcement when measuring Δ*so*-cheater frequency in mycelium and spores" in Supplementary Discussion).

qPCRs on gDNA were performed to determine the frequency of Δ*so* and wild-type nuclei within the heterokaryotic mycelium and derived spores. The principle was to differentially quantify the *hygB* gene (i.e., Δ*so* nuclei) and *so* gene (i.e., wild-type nuclei) in a gDNA sample derived from a heterokaryon between the Δ*so* cheater and the wild type. Each qPCR reaction (8 μl) was run in six technical replicates and per reaction contained: 4 μl 2× iQ SYBR Green SuperMix (Bio-Rad), 0.16 μl 10 μM of each primer, 3 μl of gDNA (18–990 pg), 0.68 μl sterile MQ-water. To amplify the *hygB* gene, we used hygB_2f/hygB_2r primer pair, whereas for amplifying the *so* gene, we used so_1f/so_1r primer pair (Supplementary Table 3). The amplification program was set to 3 min 95 °C, followed by 40 cycles of 30 s at 95 °C and 30 s at 60 °C. The reactions were run in a Bio-Rad CFX96 thermocycler. The qPCR results were analysed in Bio-Rad CFX Manager v.2.0 software. Baseline threshold line was set arbitrarily at the exponential phase of PCR (500 relative fluorescence units). Six technical replicates were averaged and Ct difference was calculated, based on which the frequency of each allele could be determined, as follows. For example, if there is no difference (ΔCt = 0) in amplification between *so* and *hygB* amplifications in a given DNA sample, this would mean equal frequencies (50/50%) of Δ*so* and wild type. Analogously, if we see a difference of 1 cycle (ΔCt = 1) between *so* and *hygB* amplifications, this would mean that there is twice more of one genotype than the other (i.e., 66/33% ratio). Continuing this logic, ΔCt = 2 would give 80/20% ratio, and so on. Primer efficiencies and the overall method principle were verified in pilot qPCR runs on gDNA samples containing known ratios of each genotype.

For the statistics, we used linear regression and one-tailed Student's *t*-test to see whether the differences in Δ*so* dynamics between the end and the start of the heterokaryotic mycelium and between the spores and mycelium are significant, with the following results:

#linear regression in enforced conditions (without inositol and pantothenic acid supplementation) after linear mycelial growth (end/start mycelial ratio of Δ*so*). *F*-statistic 1,8 = 0.5143, df = 8, Multiple *R*-squared: 0.0604, *P* = 0.4937;

#linear regression in non-enforced conditions (with inositol and pantothenic acid supplementation) after linear mycelial growth (end/start mycelial ratio of Δ*so*). *F*-statistic 1,8 = 5.309, df = 8, Multiple *R*-squared: 0.3989, *P* = 0.05014;

#one-sample one-tailed Student's *t*-test (we are specifically interested if it is less than 1) in enforced conditions (without inositol and pantothenic acid

supplementation) after linear mycelial growth (end/start mycelial ratio of $\Delta so$). Mean = 0.8576329, $t = -3.6702$, df = 9, $P = 0.002577$;

#one-sample one-tailed Student's $t$-test (we are specifically interested if it is less than 1) in non-enforced conditions (with inositol and pantothenic acid supplementation) after linear mycelial growth (end/start mycelial ratio of $\Delta so$). Mean = 0.6131703, $t = -11.692$, df = 9, $P = 4.803e-07$;

#linear regression in enforced conditions (without inositol and pantothenic acid supplementation) after sporulation of mycelium (spores/mycelium ratio of $\Delta so$). $F$-statistic 1,18 = 52.97, df = 18, Multiple $R$-squared: 0.7464, $P = 9.176e-07$;

#linear regression in non-enforced conditions (with inositol and pantothenic acid supplementation) after sporulation of mycelium (spores/mycelium ratio of $\Delta so$). $F$-statistic 1,18 = 15.71, df = 18, Multiple $R$-squared: 0.466, $P = 0.0009106$.

**Competitions of $\Delta so$ against the vegetatively incompatible strains**. To see whether fusion is required for the cheating, the $\Delta so$ mutant (*het* Cde) was competed against vegetatively incompatible strains, including vegetatively compatible (*het* Cde) wild type strain as positive control. The vegetatively incompatible strains carried a different set of *het* genes (cDE, CdE, cde) which would block the successful fusion, as only the same suit of *het* alleles would allow this. The competitions were set up as explained before (see section "Competition assays") at two starting ratios—10 and 90%, in triplicates. After the competition, appropriate dilutions of the collected spores were plated counting plates for phenotyping, as well as frozen for the subsequent gDNA extraction qPCR analyses, as explained before (see section "Frequency of $\Delta so$ cheater in heterokaryons"). The results of the two methods (phenotyping and qPCR) were consistent (Supplementary Fig. 8); the results obtained with the qPCR are presented in the main text of this manuscript (Fig. 2b).

**Transfer experiments**. To track the dynamics of $\Delta so$ in the mixture with the wild type, the transfer experiments were performed as follows. The strains (*inl* or *pan* deficient) were pre-grown in VMM slants for 6–7 days, and the spores were collected with sterile MQ-water and brought to $4 \times 10^7$ sp/ml. The spores of $\Delta so$ were mixed with wild-type spores in 1:9 ratio and 50 µl of the mixes were mat-inoculated on *inl* and *pan* supplemented VMM slants, as before (see section "Competition assays"). After 4 days, the spores were washed with 5 ml MQ-water, the total spore yield was determined, 1% of the total spores (50 µl) was transferred to a fresh VMM (inositol and pantothenic acid supplemented) slant. At the same time, an appropriate dilution of those spores was inoculated on three types of counting plates (four plates with inositol; four plates with pantothenic acid; four plates without supplementation) to determine the frequency of each genotype, including heterokaryotic spores. The process was repeated every 4 days for four transfers in triplicates. $\Delta so$-inl/$\Delta so$-pan (1:1 starting ratio) transfers were done in parallel as a control to check for the degree of heterokaryon formation between $\Delta so$ and $\Delta so$. To account for possible marker effects, the $\Delta so$/wt transfer experiments were also done with reciprocal marker combinations. Additional control experiments were run using reciprocally labeled wild types, with starting frequencies of ~10% in each of the two combinations (wt-inl/wt-pan).

**Statistical analyses and graphs**. Statistical analyses were performed in R 3.6.1[55]. The graphs were made using ggplot2 R package[56] and esthetics edited in Adobe Illustrator CC 2018.

**Statistics and reproducibility**. All experiments were performed once, except pairwise competitions of $\Delta so$ cheater with the wild type. The repetition of this experiment included a narrower range of starting frequencies of $\Delta so$ cheater, yielding similar results.

**Reporting summary**. Further information on research design is available in the Nature Research Reporting Summary linked to this article.

## Data availability

Raw sequencing data of *Neurospora crassa* morphotypes have been deposited in the GenBank database under BioProject accession PRJNA605104. Unique identifiers for the databases were obtained from Resource Identification Portal (RRID:SCR_008143, RRID:SCR_001372, RRID:SCR_006013. Fungal Genetics Stock Center database was used to search for knockout mutants for this study [http://www.fgsc.net/fgsc/search_form.php]. Gene nomenclature was followed from FungiDB. Source data are provided with this paper.

## Code availability

Computer code used in the analysis is available at Github[57]: [https://github.com/ecoevomicsjoost/neurospora_pipeline].

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

## Acknowledgements

We thank Jacobus Boomsma, Ben Auxier, Bas Zwaan for helpful comments on an earlier version of this manuscript, Arjan de Visser for helpful discussion, and Marc Maas for making Fig. 4. D.K.A., E.B., and A.A.G.-G. were supported by the Netherlands Organisation for Scientific Research (VICI; NWO 86514007).

## Author contributions

D.K.A., A.A.G.-G., A.J.M.D., and E.B. developed the concept. A.A.G.-G., E.B., and C.B.M. performed the experiments. J.vdH. developed the bioinformatics pipeline. A.A.G.-G. analysed the genomes. D.K.A. and A.A.G.-G. wrote the manuscript with contributions from E.B. and A.J.M.D.

## Competing interests

The authors declare no competing interests.
