## [Peer Review File · Nature Communications]

Reviewers' Comments:

Reviewer #1:

Remarks to the Author:

Review of "Somatic deficiency causes reproductive parasitism in a fungus". This study began as an offshoot of a previous study on the evolution of cheater nuclei in a syncytial, multinucleate organism. Sequencing of the genomes of these homokaryotic cheater strains revealed they were enriched for a mutation in a gene, called *soft*, which is required for somatic cell fusion in a number of filamentous fungi. This result is very unexpected and interesting. In this study, the authors investigate linkage between the ability to undergo fusion (and form a heterokaryon) and nuclear cheating by *soft*. Although the data are well presented, there is not enough information in the figure legends to understand the experiments without perusing the supplemental information. Additionally, the role of fusion in cheating is over-stated, as nuclei containing mutations in other genes that are required for fusion do not necessarily cheat. This issue should be addressed in the discussion.

Comments:

1. Are the strains constructed for this study submitted to a public repository?
2. The authors indicate that *soft* mutant nuclei become enriched in conidia and, because the *soft* mutation is recessive, that many of the conidia are homokaryotic for the *soft* mutation. These results suggest that sectoring in the colony occurs due to lack of fusion, so that conidia/conidiophores originate from regions with only *soft* mutant nuclei. It would be helpful to know if sectoring also occurs in wild type nuclei, for example, in the absence of selection for a heterokaryon (that was first established using selection), how often are conidia heterokaryotic versus homokaryotic? Including this experiment would test the sectoring hypothesis (Fig. 2 and supplemental figures).
3. The *ham-8* mutation was identified by sequencing of the evolved cheater lines and yet, the *ham-8* mutant nuclei were not enriched in competitions (supplementary info), suggesting that another mutation in the background was responsible for the cheater phenotype. This discrepancy should be addressed.
4. If fusion is the most important character determining whether a mutant nucleus can cheat or not, why were many of the fusion mutants (*mak-1*, *ham-3*, *ham-8* and *ham-4*) not enriched in the competition assays? Does that say something about the predicted function of *soft* and whether it may affect whether nuclei are capable of cheating or not? This issue needs to be addressed directly in the discussion. The text in lines 89-92 should be toned down—one would expect all fusion mutants to behave similarly if fusion was the only important character and that is not the case. The four mutants that did not show cheating should be stated explicitly in the main text, because *ham-8* mutant was not a cheater. As an aside, there are many more fusion mutants that could be tested—this might be out of scope of the study, but the conclusions on the role of fusion in cheating are currently over-stated with the data provided.
5. The figure legends in both the main text (Fig. 1, Fig. 2, Fig. 3) and in the supplemental text do not include enough information to evaluate how the experiments were done. This important aspect needs to be fixed so that the reader can evaluate the data presented in the figures themselves, without having to peruse the supplemental methods to understand the assays.
6. For the model, if WT can still fuse with *soft* mutants, how would only *soft* mutant sectors be generated? Roper et al., showed that branching is more important than fusion in accomplishing mixing of nuclei and that genetic diversity is maintained in hyphal networks by mixing flows of nuclei at all length scales. How can data from this previous study be reconciled with this one?
7. Fig 2 need to have WT inl + WT pan as a control—inoculum-less strains can undergo have nuclear ratio distortions: <http://www.fgsc.net/fgn/nn7/7roess.pdf>
8. Not clear what is meant by "enforcing" in the main text and figure legends. Be specific so that the reader can understand the experimental set up and thus the data. Otherwise, the data are opaque.
9. In the discussion, be careful about making assumptions about *Neurospora* ecology—there are still a lot of unanswered questions about how it gets to and colonizes burn sites.
10. Primers to what genes for qPCR (include in legend)—what controls were used for normalization?
11. Supp Fig. 2—have no idea from legend what was done.

12. Supp Fig. 4—add mixed heterokaryon plate so that the reader can evaluate how easy to differentiate colony morphology for the counting experiments.
13. Supp Fig. 5—not clear how data were obtained--provide more info in figure legend; same for Supp Fig. 7.
14. Supp Fig.8, define what is meant/done for “enforce” or “not enforce” heterokaryon formation.
15. The authors use *inl* and *pan* as markers, which are required in small amounts—could there be cross-feeding between hyphae in a colony and thus not really sectoring? Is it possible that the soft mutation results in leakiness? It would be useful to compare other more robust auxotrophic markers to make sure the *inl* and *pan* are not affecting the results.
16. Main text
 - a. line 31, some additional refs would be useful here.
 - b. Line 36, many more recent refs on somatic cell fusion in filamentous fungi.
 - c. Line 178, be specific about what is meant by “millennia”
17. Computer code should be deposited in GitHub, or another repository.

Reviewer #2:

Remarks to the Author:

This paper identifies the genetic basis of a somatic cheater mutation in the mold *Neurospora crassa*. It further goes on to understand the dynamics of the parasitism, identifying frequency dependence with wild type nuclei as fundamental to predicting success of the cheater alleles. The cheater alleles show remarkable parallel evolution in fusion genes, and the authors determine that fusion of the mutants is unidirectional, in that they cannot initiate fusion but can be fused to from wildtype. Overall, the authors convinced me that their system does show somatic parasitism. This is a really interesting finding when paired with the strong parallel evolution of fusion gene mutants. If this result is so easy to obtain in the lab with a model fungus, it has important implications for the likelihood that mycelial fungi live and evolve to cope with constant pressure of cheating nuclei. The paper is cleanly and clearly written which was much appreciated by this reviewer.

I really only have minor comments for the authors consideration:

My main suggestion is to better contextualize the study by making the results of the previous Bastiaans 2016 study more clear. Specifically, in trying to understand the paper, I think it would be helpful to understand how the cheaters were identified in the experimental evolution study before line 54 in which the underlying genetics of them are discussed. The previous study is really fundamental to the present and so more details should be presented. What is the mechanistic basis of the morphological variation between colonies, if known?

Do these cheater variants arise in multiple wildtype genetic backgrounds? This is such an interesting phenomenon, I would like to know if it is a general result of the species or less interesting if it was strain background specific. It seems like the morphological variants would have been noticed after long term transfer by the members of the *Neurospora* community.

The term “mobilization of resources” is used. Yet, the environment in which the cultures are grown is homogenous. Perhaps mobilization is not really what is going on here. It would be really interesting, though outside of the scope of the study to see what the hyphal morphology is associated with the $\Delta so/wt$ chimeras in comparison to wild type.

Figure 3A. Why are low starting frequencies of Δso not included?

“Consequently, less-well connected cheater-rich hyphae have a higher probability to become aerial hyphae and to contribute to spore formation.” I think the hypothesis is reasonable. However, cheaters being less connected could suggest they have lower access to resources, and you might assume they would be less able to reproduce. Alternatively, it seems possible to me that they somehow become

genetically reprogrammed to reproduce over vegetative growth than wildtype. Because they are so poor at growth and sporulation on their own, they still need the social parasitism to realize this advantage.

“exploited to a degree set by the environmental conditions such as extrinsic mortality” This sentence sounds really cool, but I really don’t understand. Is extrinsic mortality an environmental condition? Please parse this out into more sentences so we can get the idea.

Computer code should be readily available somewhere online. This is a better, though not obligatory, standard.

Reviewer #3:

Remarks to the Author:

The study is a mechanistically interesting novel addition to our understanding of the diversity of forms cheating and cheater limitation across biological systems and as such will interest many in the field of social evolution as well as fungal biologists. The authors find that a common genetic basis of nuclear cheating on *Neurospora* spore formation within aerial hyphae is mutation of the *so* gene. Isogenic *so* mutants appear to avoid the costs involved with initiation of hyphal fusion without causing a defect in the ability to respond to fusion initiated by fusion-capable hyphae. They further show that cheating nuclei appear to limit themselves both due to loss of relative advantage above a threshold frequency and negative effects on yield generated as the cheaters increase in frequency. They nicely show the centrality of fusion to cheater exploitation in experiments with fusion-incompatible strains. The experiments appear to have been well designed and conducted and support the major conclusions. The manuscript is generally well written, although I think some important semantic points need to be reconsidered or better clarified.

My primary questions and comments regard semantics. I find the distinction between soma and germline unclear (lines 37-39) and not entirely convincing. Can’t the mycelium generate new offspring without spores? If so, why are only aerial hyphae considered to be reproductive germline? Since the authors seem to make this distinction an important aspect of the narrative, including the title phrasing, it seems important to clarify it better. Why is it even necessary to refer to soma and germline to communicate this story? Why not just refer to spores vs mycelial cells? Related to this, if mycelia can reproduce as well as spores, wouldn’t it be more appropriate to refer to sporulation parasitism in the title rather than more generically to reproductive parasitism?

Minor comments (line numbers for the relevant text are given)

11. The meaning of ‘dishonest’ here isn’t clear. In the authors’ understanding of exploitation, is there an honest form of exploitation that the authors wish to distinguish from?

16-17. The theme of germline senescence seems to be addressed only in the abstract.

37. Do only aerial hyphae produce spores, or can some also form in the mycelial network? If the former, it would be helpful for non-fungal researchers to state this explicitly.

35. From this paragraph it seems that there are two kinds of hyphae - mycelial and aerial. It could be helpful to explicitly distinguish between them early for non-fungal researchers.

64. germline spelling

150. The authors might specify that it is an altruistic trait when fusion-capable strains initiate fusion with fusion-deficient strains, in that the fusion benefits the latter at a relative cost to the former.

Wouldn't it be a mutually beneficial trait when both hyphae initiate fusion? Or does only one hyphae initiate fusion even when both of two meeting hyphae are capable of doing so?

REVIEWER COMMENTS

Reviewer #1 (Remarks to the Author):

Review of “Somatic deficiency causes reproductive parasitism in a fungus”. This study began as an offshoot of a previous study on the evolution of cheater nuclei in a syncytial, multinucleate organism. Sequencing of the genomes of these homokaryotic cheater strains revealed they were enriched for a mutation in a gene, called soft, which is required for somatic cell fusion in a number of filamentous fungi. This result is very unexpected and interesting. In this study, the authors investigate linkage between the ability to undergo fusion (and form a heterokaryon) and nuclear cheating by soft. Although the data are well presented, there is not enough information in the figure legends to understand the experiments without perusing the supplemental information. Additionally, the role of fusion in cheating is overstated, as nuclei containing mutations in other genes that are required for fusion do not necessarily cheat. This issue should be addressed in the discussion.

We thank the reviewer for the constructive comments. We agree that it is good to discuss more extensively that indeed not all fusion mutations tested showed the cheating phenotype, and why this may be the case for some but not for all.

Comments:

1. Are the strains constructed for this study submitted to a public repository?

No. To test the effect of the soft mutation and other fusion mutants, we used single-gene knockout mutants publicly available from the Fungal Genetics Stock Center (FGSC). The extra five mutants constructed for this study were made by sexual crosses between strains available at FGSC. Such sexual crosses can be easily performed using standard Neurospora protocols and the phenotypes are easy to select from the progeny (flat colony morphology of fusion mutants, mating type, auxotrophy). Therefore, we don't feel the necessity to deposit our strains to FGSC.

2. The authors indicate that soft mutant nuclei become enriched in conidia and, because the soft mutation is recessive, that many of the conidia are homokaryotic for the soft mutation. These results suggest that sectoring in the colony occurs due to lack of fusion, so that conidia/conidiophores originate from regions with only soft mutant nuclei. It would be helpful to know if sectoring also occurs in wild type nuclei, for example, in the absence of selection for a heterokaryon (that was first established using selection), how often are conidia heterokaryotic versus homokaryotic? Including this experiment would test the sectoring hypothesis (Fig. 2 and supplemental figures).

We agree that a wild-type – wild-type control gives a better insight in the interaction between soft and wt. We therefore did the transfer experiment with two wild-type strains and followed heterokaryon formation over time (see Figure under point 7). A comparison with the data provided in Figure 2a of the manuscript shows that there is a difference in the rate of heterokaryon formation and the fraction of heterokaryons between a soft-wild-type culture and a wild-type – wild-type culture. Assuming random packaging of the nuclei in the spores, and an average of 2.5 nuclei per spore, we calculated the expected frequency of heterokaryotic spores based on the measured frequency of the two nuclear types, and compared those with the observed frequencies at each transfer (see below). This showed that there was a consistent underrepresentation of heterokaryotic spores, refuting complete mixing of nuclei, and that this difference generally was bigger for the soft – wild-type cultures (and - not surprisingly- biggest for the soft – soft cultures). This shows that there is less complete

mixing in the soft – wild-type cultures, consistent with reduced fusion, than in the wild-type – wild-type cultures.

3. The ham-8 mutation was identified by sequencing of the evolved cheater lines and yet, the ham-8 mutant nuclei were not enriched in competitions (supplementary info), suggesting that another mutation in the background was responsible for the cheater phenotype. This discrepancy should be addressed.

Yes, we followed this suggestion and discuss this discrepancy. We should emphasize that the ham-8 mutant had equal competitiveness relative to wild type at the tested frequency despite reduced spore production in monoculture. In addition to the possibility of epistatic interaction with a different acquired mutation, the benefit of some mutations may be at a lower frequency than the tested frequency of ~10% (more precisely, it was actually close to 14% in ham-8 competition, as evidenced by plating the starting competition mixture). In the evolution line the cheater (16t2) reached a frequency of 15% (which was the lowest cheater frequency of the eight low-relatedness lines after 31 generations), suggesting that 15% is indeed close to the frequency where competitive success is equal to the wild-type ancestor.

4. If fusion is the most important character determining whether a mutant nucleus can cheat or not, why were many of the fusion mutants (mak-1, ham-3, ham-8 and ham-4) not enriched in the competition assays? Does that say something about the predicted function of soft and whether it may affect whether nuclei are capable of cheating or not? This issue needs to be addressed directly in the discussion.

This is a valid point, and we followed this suggestion. One possibility is that the frequency dependence differs between different fusion mutants, and that some manifest cheating phenotypes at lower frequencies than the tested ~10%.

The text in lines 89-92 should be toned down—one would expect all fusion mutants to behave similarly if fusion was the only important character and that is not the case. The four mutants that did not show cheating should be stated explicitly in the main text, because ham-8 mutant was not a cheater.

Of the seven tested fusion mutants (additional to soft), four displayed the cheating behaviour, so five of the eight tested fusion mutants displayed the cheating behaviour. Following this suggestion, we now explicitly mention the four, and added a sentence that fusion ability per se is not the entire explanation for the cheating phenotype.

As an aside, there are many more fusion mutants that could be tested—this might be out of scope of the study, but the conclusions on the role of fusion in cheating are currently overstated with the data provided.

Indeed, testing many more fusion mutants is out of scope of the present study. But, as also explained in the previous answers, we added qualifying discussion on the relevance of fusion and mention the results not consistent with the role of fusion.

5. The figure legends in both the main text (Fig. 1, Fig. 2, Fig. 3) and in the supplemental text do not include enough information to evaluate how the experiments were done. This important aspect needs to be fixed so that the reader can evaluate the data presented in the figures themselves, without having to peruse the supplemental methods to understand the assays.

We followed this suggestion and added extra details to the figure captions.

6. For the model, if WT can still fuse with soft mutants, how would only soft mutant sectors be generated? Roper et al., showed that branching is more important than fusion in accomplishing mixing of nuclei and that genetic diversity is maintained in hyphal networks by mixing flows of nuclei at all length scales. How can data from this previous study be reconciled with this one?

This is a good point, which we now discuss. First of all, a crucial assumption of our model is that fusion between wild type and soft is highly reduced relative to fusion between wild type with wild type. This explains why the selection of soft is frequency dependent: i) at a high frequency of wild type, the wild type sub-colonies will be highly connected, and each soft-sub-colony will be maximally surrounded by wild type, giving a high probability that it will fuse with at least one of the wild-type hyphae around it; ii) conversely, at low frequencies of wild type, wild-type sub-colonies will often be broken up by soft patches since fusion between wild type and soft is rare and soft sub-colonies will be more often neighbored by soft sub-colonies and less often by wild-type sub-colonies. This means that wild-type sub-colonies will more often not fuse with soft, meaning that they are not parasitized and so have a relative benefit over unfused soft sub-colonies.

What we do not really know is how much mixing there is upon fusion. We do know, however, that starting with a heterokaryon, the soft mutant has a relative benefit to reach the spores, which also explains its initial selection (as it presumably will find itself in a heterokaryon). Our model assumes that there is sufficient sectoring between soft-rich and wild-type rich hyphae in the heterokaryotic mycelium. Part of this structuring is because of incomplete mixing upon fusion between homokaryons, and additionally there may be sectoring within heterokaryons. Indeed, Roper et al. found that there was extensive nuclear migration, both in soft and in wild-type mycelia and they also found that branching topology of soft mycelia was optimal for cytoplasmic mixing in soft. Nevertheless, they showed that a heterokaryotic colony lacking hyphal fusion had much more segregation than a wild-type heterokaryon.

7. Fig 2 need to have WT inl + WT pan as a control—inositol-less strains can undergo have nuclear ratio distortions: <http://www.fgsc.net/fgn/nn7/7roess.pdf>

We performed the suggested control experiments and indeed, there was a strong marker effect for the inositol deficient strain (see below). We added those data as a Supplemental Figure 7. Interestingly, despite this marker effect, however, soft increased its frequency in competition with wild type (Figure 2a, middle panel). It should be pointed out that those transfer experiments were all done without enforcement of the heterokaryon. We thank the reviewer for the suggestion to do these control experiments and pointing out to the possible marker effect.

8. Not clear what is meant by “enforcing” in the main text and figure legends. Be specific so that the reader can understand the experimental set up and thus the data. Otherwise, the data are opaque.

By ‘enforcing’ we mean non-supplemented conditions, so that only heterokaryons can grow.

9. In the discussion, be careful about making assumptions about *Neurospora* ecology—there are still a lot of unanswered questions about how it gets to and colonizes burn sites.

*We agree, and added that the natural ecology of *N. crassa* is not well known and added some further details.*

10. Primers to what genes for qPCR (include in legend)—what controls were used for normalization?

*The primers were designed to amplify the *hygB* gene (for Δso nuclei) and the *so* gene (for wild-type nuclei). The elegance of this method is that the normalization is not needed, because the principle is to look at cycle (*Ct*) difference between *so* and *hygB* amplifications in the same DNA sample. For example, if we see no difference in *Ct* between *so* and *hygB* amplifications, this would mean equal frequencies (50%/50%) of Δso and wild type in a given DNA sample. Analogously, if we see a difference of 1 cycle (*Ct*=1) between *so* and *hygB* amplifications, this would mean that there is twice more of one genotype than the other (i.e. 66%/33% ratio). Continuing this logic, if there is *Ct*=2, this would mean 4 times difference between the two genotypes (i.e. 80%/20% ratio), and so on. As we mentioned in the Supplementary Information, we confirmed primer efficiencies to be optimal, and also the overall principle of this method was validated on DNA samples with known ratios of both*

genotypes. However, to provide extra details on the principle, we added a couple of sentences in the Supplementary Information to clarify this.

11. Supp Fig. 2—have no idea from legend what was done.

To follow the competitive benefit of Δso in time, we sampled the competitions every day for six days. Extra details on this were added to the figure legend.

12. Supp Fig. 4—add mixed heterokaryon plate so that the reader can evaluate how easy to differentiate colony morphology for the counting experiments.

We followed this suggestion and added a plate picture with a mixture of soft and wild-type colonies (see updated Supplementary Fig. 4).

13. Supp Fig. 5—not clear how data were obtained--provide more info in figure legend; same for Supp Fig. 7.

We added extra details to the figure captions.

14. Supp Fig.8, define what is meant/done for “enforce” or “not enforce” heterokaryon formation.

By ‘enforce’ we mean non-supplemented medium, so that only heterokaryons can grow. Conversely, by ‘not enforce’ we mean supplemented conditions, so that both the two homokaryons and the heterokaryons can grow.

15. The authors use *inl* and *pan* as markers, which are required in small amounts—could there be cross-feeding between hyphae in a colony and thus not really sectoring? Is it possible that the soft mutation results in leakiness? It would be useful to compare other more robust auxotrophic markers to make sure the *inl* and *pan* are not affecting the results.

We can exclude this possibility for two reasons. First, most of the experiments used complete medium, so a medium not enforcing the heterokaryotic condition. Thus, any cross feeding is irrelevant in this period. Second, to determine the frequencies, we used three media including a minimal medium to determine the frequency of heterokaryotic spores. As the third panel of Figure 2a shows, for the two soft strains, we found virtually no growth on the non-supplemented plates, thus refuting that cross feeding between hyphae is responsible for the results of counted heterokaryotic spores (which would be highly unlikely for yet another reason, since spores were inoculated at low density to enable counting on the counting plates).

16. Main text

a. line 31, some additional refs would be useful here.

We followed this suggestion and added three additional references to work on sessile invertebrates.

b. Line 36, many more recent refs on somatic cell fusion in filamentous fungi.

We have added two references to recent reviews.

c. Line 178, be specific about what is meant by “millennia”

We changed this to ‘with an estimated age of at least 2,500 years.’

17. Computer code should be deposited in GitHub, or another repository.

We deposited the code in GitHub (https://github.com/ecoevomicsjoost/neurospora_pipeline)

Reviewer #2 (Remarks to the Author):

This paper identifies the genetic basis of a somatic cheater mutation in the mold *Neurospora crassa*. It further goes on to understand the dynamics of the parasitism, identifying frequency dependence with wild type nuclei as fundamental to predicting success of the cheater alleles. The cheater alleles show remarkable parallel evolution in fusion genes, and the authors determine that fusion of the mutants is unidirectional, in that they cannot initiate fusion but can be fused to from wildtype. Overall, the authors convinced me that their system does show somatic parasitism. This is a really interesting finding when paired with the strong parallel evolution of fusion gene mutants. If this result is so easy to obtain in the lab with a model fungus, it has important implications for the likelihood that mycelial fungi live and evolve to cope with constant pressure of cheating nuclei. The paper is cleanly and clearly written which was much appreciated by this reviewer.

We thank the reviewer for the positive comments.

I really only have minor comments for the authors consideration:

My main suggestion is to better contextualize the study by making the results of the previous Bastiaans 2016 study more clear. Specifically, in trying to understand the paper, I think it would be helpful to understand how the cheaters were identified in the experimental evolution study before line 54 in which the underlying genetics of them are discussed. The previous study is really fundamental to the present and so more details should be presented. What is the mechanistic basis of the morphological variation between colonies, if known?

We followed this suggestion and provided more information in the revised manuscript on how they could be recognized.

Do these cheater variants arise in multiple wildtype genetic backgrounds? This is such an

interesting phenomenon, I would like to know if it is a general result of the species or less interesting if it was strain background specific. It seems like the morphological variants would have been noticed after long term transfer by the members of the Neurospora community.

We discuss earlier observations that are consistent with our finding (reference 29). Another possibility is that such morphological “defective” mutants can be simply ignored and/or discarded, as this is described in a laboratory manual for another fungus – Fusarium (“The Fusarium Laboratory Manual” First edition, 2006, section “Degenerate cultural variants” p. 23). The manual states that “The use of standard culturing procedures enables degenerate cultural variants to be recognized quickly and discarded”. Interestingly, they describe sporeless and female-sterile morphotypes that may arise (similar to our soft mutants), but “little is known of the mechanisms that underlie the process”. Well, our paper provides one explanation and indicates the generality of cheater threat. We added this curious detail to our discussion in the paper.

The term “mobilization of resources” is used. Yet, the environment in which the cultures are grown is homogenous. Perhaps mobilization is not really what is going on here. It would be really interesting, though outside of the scope of the study to see what the hyphal morphology is associated with the Δ so/wt chimeras in comparison to wild type.

Yes, we agree, but indeed outside of the scope of the present manuscript.

Figure 3A. Why are low starting frequencies of Δ so not included?

Because it is not possible to manipulate the frequency of a heterokaryon, we randomly selected heterokaryotic mycelia, and there happened to be no samples with very low starting frequencies of soft. After mycelial growth, however, the frequency had dropped, which explains that we have lower starting frequencies in 3B, since we used both the mycelia from the start of the race tube and the end of the race tube (this also explains that 3B has twice as many data points as 3A).

“Consequently, less-well connected cheater-rich hyphae have a higher probability to become aerial hyphae and to contribute to spore formation.” I think the hypothesis is reasonable. However, cheaters being less connected could suggest they have lower access to resources, and you might assume they would be less able to reproduce. Alternatively, it seems possible to me that they somehow become genetically reprogrammed to reproduce over vegetative growth than wildtype. Because they are so poor at growth and sporulation on their own, they still need the social parasitism to realize this advantage.

This is an interesting suggestion. A somewhat related possibility is that sporulation is triggered when food becomes exhausted. Since this will happen earlier in soft-rich hyphae, their benefit may be based on the timing of reproduction. We tested and refuted this hypothesis, as explained in the Supplemental Discussion (first paragraph, and Suppl. Fig.2).

“exploited to a degree set by the environmental conditions such as extrinsic mortality” This sentence sounds really cool, but I really don’t understand. Is extrinsic mortality an environmental condition? Please parse this out into more sentences so we can get the idea.

The full sentence was: “Our experiments demonstrate that fusion is best conceptualized as an altruistic trait that can be exploited to a degree set by the environmental conditions such as extrinsic mortality, which in turn will likely select for defense mechanisms.” We modified this to: “Our experiments demonstrate that fusion is best conceptualized as an altruistic trait that can be exploited. However, the extent of exploitation will depend on the degree to which asexual spores serve local dispersal, which in turn depends on environmental conditions. Furthermore, the ecology may or may not provide extrinsic barriers to the mycelial lifespan,

thus limiting the time span for cheater mutations to arise in a single clone, thereby influencing the selective pressure for defense mechanisms.”.

Computer code should be readily available somewhere online. This is a better, though not obligatory, standard.

We deposited the code in GitHub (https://github.com/ecoenvomicsjoost/neurospora_pipeline) (was also suggested by Reviewer 1).

Reviewer #3 (Remarks to the Author):

The study is a mechanistically interesting novel addition to our understanding of the diversity of forms cheating and cheater limitation across biological systems and as such will interest many in the field of social evolution as well as fungal biologists. The authors find that a common genetic basis of nuclear cheating on *Neurospora* spore formation within aerial hyphae is mutation of the *so* gene. Isogenic *so* mutants appear to avoid the costs involved with initiation of hyphal fusion without causing a defect in the ability to respond to fusion initiated by fusion-capable hyphae. They further show that cheating nuclei appear to limit themselves both due to loss of relative advantage above a threshold frequency and negative effects on yield generated as the cheaters increase in frequency. They nicely show the centrality of fusion to cheater exploitation in experiments with fusion-incompatible strains. The experiments appear to have been well designed and conducted and support the major conclusions. The manuscript is generally well written, although I think some important semantic points need to be reconsidered or better clarified.

My primary questions and comments regard semantics. I find the distinction between soma and germline unclear (lines 37-39) and not entirely convincing. Can't the mycelium generate new offspring without spores? If so, why are only aerial hyphae considered to be reproductive germline? Since the authors seem to make this distinction an important aspect of the narrative, including the title phrasing, it seems important to clarify it better. Why is it even necessary to refer to soma and germline to communicate this story? Why not just refer to spores vs mycelial cells? Related to this, if mycelia can reproduce as well as spores, wouldn't it be more appropriate to refer to sporulation parasitism in the title rather than more generically to reproductive parasitism?

*The reviewer raises valid points, indeed, since fungi are modular organisms there is no early differentiation between germline and soma. However, in our evolution experiments, only conidia (asexual spores) were transferred, so effectively, they represent the germline in this context. As we discuss in the paper, in real life, the asexual phase of *N. crassa* may be restricted in time, since its ecology is linked to post-fire habitats, and only sexual spores survive fire. However, in the context of our evolution experiment, asexual spores are the germline, so we propose to maintain this terminology.*

Minor comments (line numbers for the relevant text are given)

11. The meaning of 'dishonest' here isn't clear. In the authors' understanding of exploitation, is there an honest form of exploitation that the authors wish to distinguish from?

This seems to be a pleonasm, since exploitation per definition is dishonest indeed! We therefore removed 'dishonest'.

16-17. The theme of germline senescence seems to be addressed only in the abstract.

We now use the same term in the discussion of the revised manuscript and discuss it in the context of senescence in general.

37. Do only aerial hyphae produce spores, or can some also form in the mycelial network? If the former, it would be helpful for non-fungal researchers to state this explicitly.

Yes, indeed, macroconidia develop exclusively from aerial hyphae (microconidia do not, but those function primarily as male fertilizing elements for crosses). We have added this to the text.

35. From this paragraph it seems that there are two kinds of hyphae - mycelial and aerial. It could be helpful to explicitly distinguish between them early for non-fungal researchers.

Aerial hyphae differentiate from the hyphal network late in development, and are thus not sequestered early. Therefore, we have changed the sentence "Such spores are produced exclusively by aerial hyphae, which can thus be considered as reproductive structures supported by a substrate-bound somatic hyphal network that acquires the organic resources for growth." to "Such spores are produced exclusively by aerial hyphae, which differentiate from the mycelium and can thus be considered as reproductive structures supported by a substrate-bound somatic hyphal network that acquires the organic resources for growth."

64. germline spelling

This was not meant to refer to germline, but to germling. As this is probably a term not known to most non-mycologists, we changed 'hyphal and germling fusion' to 'fusion between hyphae and germinating spores'

150. The authors might specify that it is an altruistic trait when fusion-capable strains initiate fusion with fusion-deficient strains, in that the fusion benefits the latter at a relative cost to the former. Wouldn't it be a mutually beneficial trait when both hyphae initiate fusion? Or does only one hyphae initiate fusion even when both of two meeting hyphae are capable of doing so?

The reviewer raises an interesting point. Yes, in principle, at the cell level, fusion is mutually beneficial. However, at the level of a colony, a fusion-deficient colony profits more from fusion than a wild-type colony. This is presumably because the fusion-deficient strain provides fewer resources to the fused colony than the fusion-proficient strain.

Reviewers' Comments:

Reviewer #1:

Remarks to the Author:

The authors have addressed my comments and suggestions in this nicely revised manuscript.

Reviewer #2:

Remarks to the Author:

I have reviewed the changes made to address the previous round of comments. I found the way they were handled to be sufficient, and I believe this rather strong paper is considerably improved. The authors are starting to link potentially some of the older literature on culture degeneration with these somatic mutations. This type of degeneration is a fascinating topic for considering how colonial organisms cope with the constant evolution of cheater cell lineages.

Reviewer #3:

Remarks to the Author:

The authors have addressed my comments and questions to my satisfaction. I congratulate them on a very nice study.